# Semantic-Aware Local Image Editing with a Single Mask Operation

## Abstract

*We introduce a user-friendly method for controllable image editing, where users simply draw an imprecise mask on the reference image to adaptively transfer its stylistic elements to the target image. Our approach, Adaptive Paste-GAN, is an optimization-based method that relies on intermediate feature maps of GANs for supervision. The method consists of two stages: ROI detection and local editing. In the ROI detection stage, deformable feature matching identifies the optimal editing region within the StyleGAN feature maps. In the editing stage, the latent code is optimized to align the target image's ROI features with those of the reference, while applying regularization to minimize changes outside the ROI. Experimental results demonstrate the precision of ROI detection and show that our method effectively balances locality and global consistency during optimization, and aligns well with user intent across various image categories. The code will be made available upon publication.*

## 1. Introduction

As modern generative models become increasingly capable of creating photorealistic images, the demand for controllable image synthesis has grown significantly to meet practical application needs. Numerous methods for controllable image generation and editing have emerged. These methods introduce control mechanisms across various dimensions, such as attribute label guidance [1, 19], semantic segmentation maps or sketches [2, 14, 26], 3D perspective manipulation [3, 10], and textual descriptions [12, 27, 29, 36]. More recently, a novel technique employing drag points for image editing was introduced in [25]. However, transferring local semantic information from a reference image to a target image offers a more nuanced form of control compared to the previously discussed approaches.

We propose a new approach for controllable image editing. Users need only draw a single mask over a semantic region in the reference image, and the model will adaptively paste the semantic information onto a corresponding area in the target image. The ideal framework for such localized,

semantically-aware image edits should possess several key properties: 1) Convenience: allowing users to select the copy area only in the reference image, while the model automatically detects the corresponding area in the target image. 2) Flexibility: enabling the copying of any user-selected region without constraints imposed by semantic segmentation. 3) Semantic Awareness: given that the user's mask may not be precise, the model should extract the semantic information rather than directly transferring pixels. For example, in Fig. 1, even with a rough mask, the model should extract the "absence of a beard" as the local semantic information. 4) Locality: the transfer should be confined to the regions of interest, ensuring minimal impact on areas outside these regions to maintain the realism and plausibility of the output. 5) Generality: the method should be applicable to various object categories without requiring specific training or labeling.

The closest existing task to this setup is image blending, but traditional image blending methods are generally neither convenient nor semantically-aware. These methods typically require finely crafted masks on both the reference and target images. Nevertheless, existing image blending methods [6, 7, 17, 18, 22, 31] achieve only a subset of the aforementioned desirable properties. Our objective is to integrate all these attributes comprehensively.

To achieve such localized, semantically-aware editing across images, two primary challenges must be addressed: ROI (Region of Interest) detection and semantically-aware local editing. Given the arbitrary nature of the user's mask and the structural variations across images, identifying the optimal target region in the target image via ROI detection is crucial. Moreover, achieving a balance between locality and semantic awareness is essential for adaptively transferring the selected stylistic elements.

We introduce Adaptive Paste-GAN, a novel approach for region transfer across images that employs a copy-paste paradigm within the framework of StyleGAN [16]. In Style-GAN, an image is generated by progressively infusing style information from a latent code into multiple upsampling synthesis blocks. Our method leverages only the intermediate features from both the reference and target images within

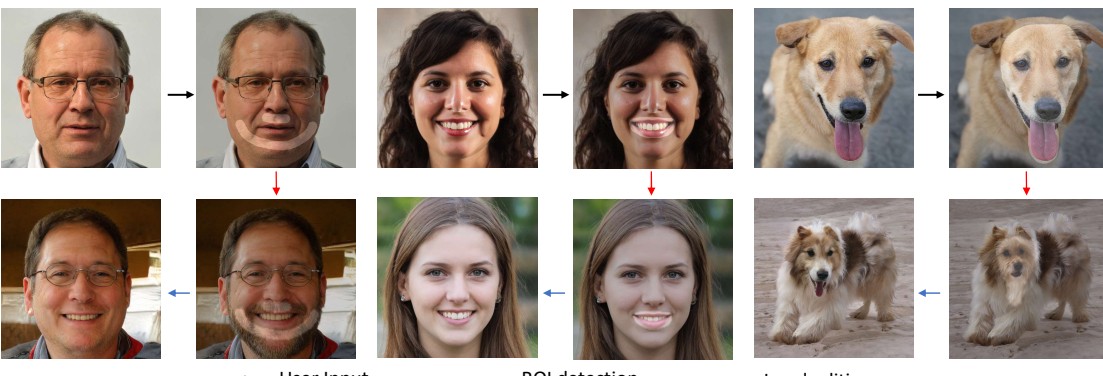

Figure 1. Our technique enables context-aware element transplantation between GAN-synthesized images. Users simply delineate a region in the reference image (black arrow), and our algorithm autonomously finds a matching area in the target image (red arrow), seamlessly integrating the local style while preserving contextual coherence (blue arrow).

these synthesis blocks and relies entirely on optimization, eliminating the need for additional training or annotations. This allows it to be applicable across various image types.

Our methodology unfolds in two stages: ROI detection and local semantically-aware editing. Specifically, ROI detection involves extracting features from the user-defined copy region in the reference image and identifying a corresponding region in the target image's feature space. The ROI can be scaled, rotated, and translated to achieve the highest cosine similarity with the reference features through deformable feature matching. During the editing phase, the latent code is optimized to make the features in the target image's ROI closely resemble those of the reference region, while a regularization loss is applied to minimize changes outside the ROI. Importantly, in the initial editing stages, some modifications are allowed outside the ROI to preserve the overall plausibility and realism of the output. As editing progresses, these external changes are minimized, enhancing the localized nature of the edit. This process enables a dynamic balance between maintaining the integrity of the entire image and focusing edits on specific areas, offering a significant improvement over the static nature of traditional image blending.

By providing real-time results within an interactive interface, users can select the level of semantic awareness or locality that aligns with their intentions. For instance, if a user desires a more semantically-aware edit, such as removing a beard in Fig. 1, the process can be halted earlier. Conversely, for more localized control, such as adjusting the mouth in Fig. 1, the optimization process can be extended. This approach not only allows for precise localized adjustments but also enables users to roughly define the copy area to capture essential semantics (e.g., beards, hair). Despite being optimization-based, Adaptive Paste-GAN remains highly efficient, thanks to its streamlined network architecture and straightforward loss calculations, typically requiring only tens of seconds of processing time on a single RTX 4090 GPU. This efficiency allows users to rapidly experiment within an interactive interface, facilitating the swift attainment of images that meet their expectations.

Our method demonstrates outstanding performance across diverse image categories, significantly improving localized edits while maintaining global consistency and realism. Experiments validate its superiority in achieving seamless integration of stylistic elements from reference to target images. Compared to existing methods, our approach ensures higher fidelity to the reference style with minimal alterations to surrounding areas, showcasing its effectiveness in precise, localized semantic transfers. Notably, our technique's unique capability for local style interpolation and attribute decoupling further underscores its advanced control and flexibility. The robustness of our model is further confirmed through sequential editing tests, highlighting its potential in practical applications without inducing image distortion or loss of feature integrity. These results collectively demonstrate the innovation and applicability of Adaptive Paste-GAN in controllable image generation and editing.

The contributions of this work can be summarized as follows:

- A novel approach to controllable image editing that adaptively transfers local semantic content from a reference image to a corresponding location in the target image using only a user-defined mask.
- Accurate ROI detection through *deformable feature matching*, achieved with minimal network optimization.
- Optimization-based local semantically-aware editing that balances semantic awareness and locality.
- Experiments demonstrate our method's convenience, flexibility, semantic awareness, locality, and generality.

## 2. Related Work

### 2.1. Controllable Image Synthesis and Editing

The field of controllable image synthesis and editing predominantly leverages GAN[9] and diffusion models[13]. We primarily examines GAN-based methodologies. These approaches are broadly categorized into conditional GANs and manipulating unconditional GANs.

Conditional GANs incorporate not just randomly sampled latent codes but also specific controlling conditions as constraints, for instance, semantic segmentation maps[20, 26, 30], sketches[14], and attribute values[19]. This strategy offers the benefit of directly manipulating certain characteristics of the generated images. However, a significant drawback is the substantial requirement for annotated data to accurately model the conditional distributions. Approaches like LinkGAN[34] and StyleMapGAN[17] map latent codes directly to pixel space during GAN training. Similarly, [24] demonstrates generating images with semantically dense correspondences among them, applicable in cross-image space or semantic editing. Nonetheless, these techniques necessitate the training of specialized models and extensive semantic segmentation data.

Various techniques exist for manipulating unconditional GANs, principally divided into global attribute modification via latent codes and localized editing through manipulation of intermediate features. Global attribute modification techniques typically employ external models or annotated datasets for guidance[1, 27]. Alternatively, some methodologies adopt an unsupervised approach to discern semantic directions within the latent space[11, 35]. Local editing techniques, in contrast, involve direct alterations to specific features within some certain layers in generator[6, 18, 31], offering simplicity and obviating the need for training, albeit with restricted editing capabilities. Our approach synthesizes the benefits of both strategies, employing intermediate features for global modifications by optimizing latent codes without requiring external supervision, thus facilitating more naturally edited effects.

### 2.2. Deformable Convolution

The geometric rigidity of standard 2D convolution kernels limits their adaptability to spatial transformations. Deformable Convolution, introduced by [8], enhances this adaptability by allowing the kernel to adjust to arbitrary spatial structures with minimal changes to conventional convolutions.

In standard convolution, given an input feature map $x$ and a convolution kernel $\mathbf{K}$, the output feature map $y$ at any given location $p_0$ is defined as

$$y(p_0) = \sum_{p_n \in \mathcal{R}} \mathbf{K}(p_n) \cdot \mathbf{x}(p_0 + p_n) \qquad (1)$$

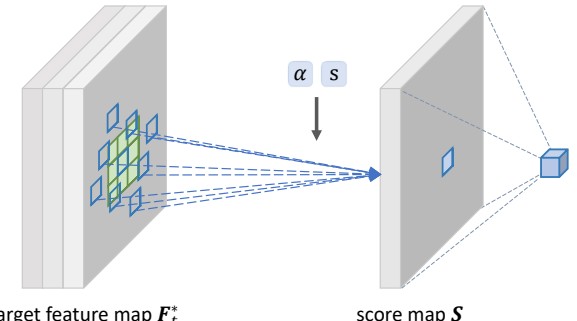

Figure 2. Deformable Feature Matching. The architecture includes a modified deformable convolutional layer, controlled by learnable parameters $\alpha$ and $s$, and a global max-pooling layer that identifies the most similar position.

Here, $p_n$ represents the enumerated locations within $\mathcal{R}$, a predefined regular grid over the input feature map $x$. For instance, a $3 \times 3$ kernel with a dilation factor of 1 defines $\mathcal{R}$ as $\{(-1, -1), (-1, 0), \ldots, (0, 1), (1, 1)\}$. Deformable convolution introduces a dynamic element to this process by incorporating an offset $\Delta p_n$ to each position $p_n$ within the grid, modifying Eq. (1) to

$$y(p_0) = \sum_{p_n \in \mathcal{R}} \mathbf{K}(p_n) \cdot \mathbf{x}(\underbrace{p_0 + p_n + \Delta p_n}_{p'_n}) \qquad (2)$$

This adaptation results in $p'_n = p_0 + p_n + \Delta p_n$ representing locations on $x$ that are irregular and often fractional. To accurately derive values for these locations, bilinear interpolation is employed. Details can be found in [8].

## 3. Method

### 3.1. Prerequisites

Our method aims to adaptively copy and paste any spatial region from a reference image to a target image within the StyleGAN framework. The StyleGAN generator includes a mapping network that transforms a Gaussian noise $\mathbf{z} \in \mathcal{N}(0, \mathbf{I})$ into a latent code $\mathbf{w} \in \mathbb{R}^{512}$, known as the $\mathbf{W}$ space. This is complemented by a synthesis network comprised of several convolutional and upsampling layers, which utilizes $\mathbf{w}$ to progressively infuse style information into a base matrix across $k$ synthesis blocks, culminating in high-resolution images. This process allows for the use of distinct latent codes at different layers, effectively creating $\mathbf{w} \in \mathbb{R}^{k \times 512}$, known as the $\mathbf{W}^+$ space, which boasts enhanced expressiveness. We represent the feature maps outputted by synthesis blocks as $\{\mathbf{F}^1, \mathbf{F}^2, ..., \mathbf{F}^k\}$, illustrating the image's features from the most coarse to the most detailed. According to [33], these features, once resized and amalgamated, serve as the input for semantic segmentation

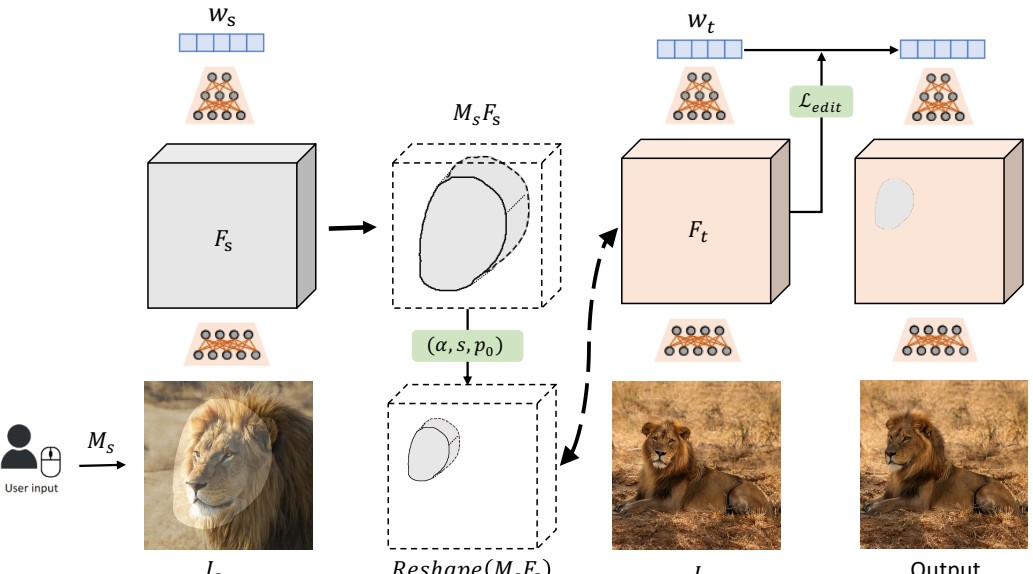

Figure 3. An overview of the local editing. Initially, ROI features are extracted utilizing the mask drawn by the user. Subsequently, the most appropriate location within the target image is identified via deformable feature matching. This location is then aligned with the target image. Finally, local editing is achieved through the optimization of latent codes.

networks, evidencing their cross-image discriminative power and semantic richness. This key insight underpins our research, demonstrating that merely manipulating these intermediate features may facilitates effective ROI detection and local editing.

### 3.2. ROI Detection

ROI detection is conceptualized as identifying the most suitable location on a target image to seamlessly integrate a local patch from a reference image, thereby maintaining the edited image's credibility and realism. The study of ROI detection is not widely explored. We hypothesize that the ideal ROI should have a semantic similarity to the local patch of the reference image. We posit that the binary mask from the reference image, denoted as $\mathbf{M}_s$, should ideally undergo transformations such as scaling, rotation, and translation when applied to the target image, without undergoing any distortion, as the shape is an important attribute to be transferred. This requirement sets it apart from tasks like semantic segmentation, object detection, or dense correspondence.

Following [33], we select $m$ semantically discriminative intermediate features $\{\mathbf{F}^{i_1}, \mathbf{F}^{i_2}, ..., \mathbf{F}^{i_m}\}$, upsample them to a consistent dimension, and concatenate to form $\mathbf{F}^*$. We introduce a technique named *deformable feature matching* for ROI detection by those features. By utilizing the features within the mask of the reference image as a convolutional kernel $\mathbf{K} = \mathbf{M}_s \cdot \mathbf{F}_s^*$, and applying this kernel over the target image's feature map $\mathbf{F}_t^*$, we can calculate the similarity at

each position.

$$S(p_0) = \sum_{p_n \in \mathcal{R}} \mathbf{K}(p_n) \cdot \mathbf{F}_t^*(p_n') \quad (3)$$

Diverging from Eq. (2), our approach ensures $p_n'$ remains semi-regular, allowing only for scalability and rotatability.

$$p_n' = p_0 + s \cdot \mathbf{T}(\alpha) \cdot p_n \quad (4)$$

Here, $s$ represents the scale factor, with $\mathbf{T}$ being the rotation matrix influenced by angle $\alpha$.

$$\mathbf{T}(\alpha) = \begin{pmatrix} \cos(\alpha) & \sin(\alpha) \\ -\sin(\alpha) & \cos(\alpha) \end{pmatrix} \quad (5)$$

The objective of the optimization process is thus defined as

$$\underset{\alpha,s,p_0}{\arg\min} -\log(S(p_0)) + \lambda_1 \cdot \mathcal{L}_{reg}(\alpha) + \lambda_2 \cdot \mathcal{L}_{reg}(s - 1/s) \quad (6)$$

Here, $\mathcal{L}_{reg}(x) = \max(0, |x| - x_{max})$ serves to regulate the rotation angle and scale ratio within permissible bounds. The implementation involves a convolution layer and a global max pooling layer, optimizing over two parameters, $s$ and $\alpha$, typically converging within tens of iterations. Finally, the mask $\mathbf{M}_t$ on the target image is obtained by simple geometric transformation.

### 3.3. Local Semantically-aware Editing

Upon obtaining the ROI, our objective is to synchronize the semantic details within the target image's $\mathbf{M}_t$ region with those of the reference image's $\mathbf{M}_s$, while ensuring the

Figure 4. Performance of our approach across multiple image categories. Reference images are shown in the first column, ROI detection results on target images are in the second column, and local editing outcomes are presented in the third column.

remainder of the image remains true to the original target. To achieve this, we initiate $\mathbf{w}$ as the latent code of the target image. We select a highly editable feature map $\mathbf{F}$ form $k$ synthesis blocks, which is bilinearly interpolated to match the dimensions of the generated image. Every feature value on $\mathbf{F}$ represents pixel-specific information across the entire image. Then, we supervise the feature map to optimize the latent code $\mathbf{w}$. The formulated supervisory loss function is as follows:

$$\mathcal{L}_{\text{edit}} = \|Reshape(\mathbf{M}_s\mathbf{F}_s) - \mathbf{M}_t\mathbf{F}\|_1$$
$$+ \gamma \|(1 - \mathbf{M}_t)\mathbf{F}_t - (1 - \mathbf{M}_t)\mathbf{F}\|_1 \quad (7)$$

Here, $Reshape(\mathbf{M}_s\mathbf{F}_s)$ involves adapting the reference image's masked feature through rotation and scaling to fit the target's $\mathbf{M}_t$ region. The first term of the equation strives for maximum coherence within the ROI to the reference, while the second term fixes the non-ROI segments. Setting $\gamma < 1$ ensures the edited image maintains overall believability and aesthetic integrity. Optimization occurs within the $\mathbf{W}^+$ space to leverage its enhanced capacity for precise edits. Despite the susceptibility of the $\mathbf{W}^+$ space to generate anomalies, our sequential editing trials confirm its robustness for this optimization task. Specifically, we focus on optimizing the semantically significant initial six layers of $\mathbf{w}$, leaving the appearance-influencing latter layers unchanged, aligning with the approach in [25].

## 4. Experiments

### 4.1. Implementation Details.

Our experiments are conducted using a single RTX 4090 GPU. For ROI detection, we use the Adam optimizer to optimize the scale factor $s$ and the angle $\alpha$ with initial step sizes of 0.5 and 5, respectively, and apply a StepLR scheduler with step size 20 and gamma 0.5 for 80 steps. To enhance optimization speed, we specifically select the third and fifth feature maps to be reshaped to $128 \times 128$ as $F^*$. During convolution, we apply dilation to ensure the size of the convolution kernel is up to $10 \times 10$. Typically, the optimization concludes within 15 seconds. For local editing, we use the Adam optimizer to optimize the latent code $w$ with a step size of 0.05. The number of steps for optimization depends on the user's trade-off between locality and global consistency. The optimization time is usually between 3 seconds and several tens of seconds.

*Datasets.* We evaluate our approach using a diverse set of pre-trained StyleGAN2 models[16] from online sources, involving datasets such as FFHQ (512)[15], LSUN Car (512)[32], AFHQCat (512) [5], and Lion (512), Dog (1024), Elephant (512) from [23]. Parenthetical numbers indicate image resolutions.

*Baseline.* For evaluating our method's proficiency in local editing, we benchmark against the feature blend approach[31], the Editing-in-Style presented in [7] and BlendNeRF[18], which likewise employs optimization. Given the novelty of our approach in ROI detection, we undertake an independent evaluation due to the absence of similar studies.

### 4.2. Evaluation of ROI Detection

Directly quantifying the efficacy of ROI detection poses a challenge, as determining the appropriateness of the detected ROI is not straightforward. Furthermore, relying on the final editing outcomes for evaluation could bias the results due to the influence of Local Editing quality. In light of these considerations, we introduce an indirect evaluation methodology. From the FFHQ-StyleGAN dataset, we randomly selected

| Landmark | Error (%) | Std Dev (%) |
|---|---|---|
| Left Eye Center | 1.63 | 0.32 |
| Right Eye Center | 1.44 | 0.34 |
| Nose Tip | 2.05 | 0.28 |
| Left Mouth Corner | 1.67 | 0.37 |
| Right Mouth Corner | 1.74 | 0.32 |

Table 1. Accuracy of ROI detection for key facial landmarks

1000 pairs of reference and target images. We assumed the user drawn mask to be a circle, with a 15-pixel radius, centered around a specific landmark on the reference facial image. The effectiveness of the ROI detection was assessed by comparing the centers of detected regions with the target images' landmarks. As illustrated in Tab. 1, the average distance across five facial landmarks by our method was below 2.05%, suggesting high accuracy in matching the landmarks between reference and target images. This indirectly demonstrates the robustness of our ROI detection technique. The practical outcomes of ROI detection are showcased in Fig. 4. Furthermore, since all editing effects presented in this paper build upon the foundation of ROI detection, the quality of these effects serves as an indirect validation of ROI detection method's efficacy.

### 4.3. Results of Editing

Fig. 4 shows our method's effects across various image categories. The second columns display ROI detection results. Despite variations in object scale and positioning, our algorithm effectively aligns with the most suitable regions through scaling and rotation. The third columns present local editing outcomes, recommended for detailed examination. Fig. 5 highlights our method's performance on facial images, with masks omitted for clarity. In experiments, masks were applied to hair, eyes, and mouth, followed by ROI detection and editing. The results demonstrate strong stylistic consistency with the reference image, while preserving the surrounding areas without visible seams. The hair transfer effect is particularly notable. Additional results are provided in the Supplementary Material.

**Real image editing.** We apply PTI inversion [28] to the real image, and then use our method to edit the real image. Fig. 6 is one of the results. It can be seen that our method can also achieve good editing effects on real images. More results can be found in the Supplementary Material.

**Comparison.** As shown in Fig. 6, we present a comparative analysis of our method against others. Despite superficial similarities with Feature blend[31], our method and it exhibit distinct outcomes. Feature blend results in images where edited regions show noticeable seams with surrounding areas. In contrast, our method, which optimizes latent codes, generates edited images that seamlessly remain within the

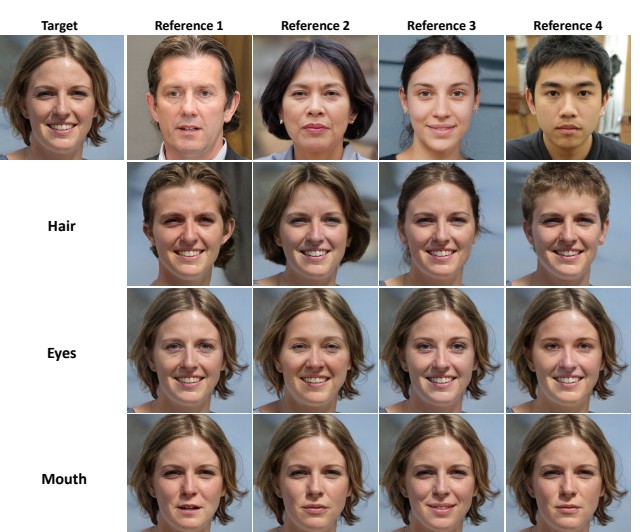

Figure 5. Our approach enables localized style transfer from any selected region (left column) of the reference image (top row) directly to the target image (top left), ensuring that the remaining areas undergo minimal and seamlessly natural alterations.

$W^+$ space, eliminating noticeable discrepancies. Although Editing-in-Style[7] produces natural-looking outcomes, its style variance, particularly the mouth, significantly deviates from the reference image. Similarly, the optimization-based BlendNeRF [18] can accurately replicate local regions, but due to using image pixels as the optimization target, even with fewer steps (50 steps), it exhibits color discrepancies (e.g., in the nose) and retains unintended details (e.g., the beard). Our method, however, ensures exceptional stylistic consistency and maintains overall color uniformity in localized edits relative to the reference image. Additionally, after mouth replacement, our model reduces the nasolabial folds in response to the mouth changes, enhancing the image's naturalness through semantic awareness.

**Locality.** The model's focus on specific regions, termed "locality", was assessed using squared-error in pixel space between target images and edited outputs. This evaluation, conducted on 50,000 FFHQ-StyleGAN samples with semantically segmented masks for eyes, noses, and mouths, calculated squared distances in the CIELAB color space. As shown in Fig. 7, our approach mainly affects areas within the masks, with minor unintended changes in adjacent regions. For example, while the mouth's segmentation excludes nasolabial folds, alterations between smiling and non-smiling expressions may cause noticeable changes in these areas. These adjustments, though outside the masked regions, highlight the model's nuanced approach to maintaining image authenticity while focusing on localized edits.

**Local Style Interpolation.** The progression of image modifications during the editing process is showcased in Fig. 8. Our technique enables the semantic interpolation of

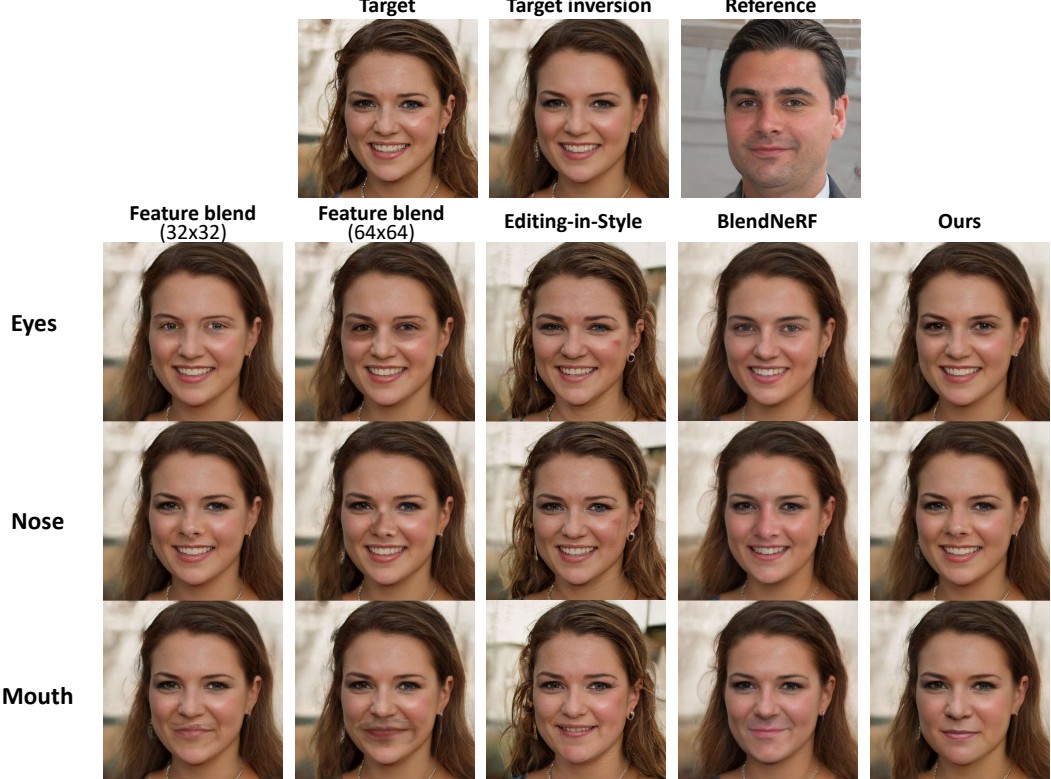

Figure 6. Comparison between our method and baseline approaches. It can be seen that our local edit has higher consistency with the reference images, and the overall output is more realistic and reasonable.

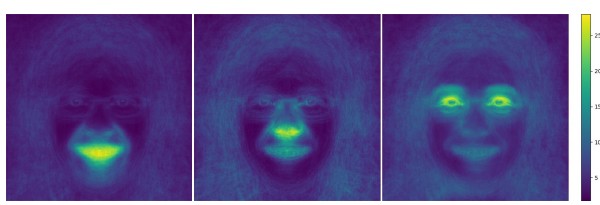

Figure 7. Mean squared error (MSE) heatmaps between the target image and the edited output in the CIELAB color space for the mouth, nose, and eyes.

style within the ROI, without altering the rest of the image. Intriguingly, as depicted in Fig. 9, this localized semantic editing also demonstrates a notable level of decoupling. The process starts with the removal of glasses, maintaining the original eye appearance, before transitioning to the reference image's eyes. This feature is key to matching user intent, allowing users to stop the process at their desired point to achieve a satisfactory result.

**Multiple Editing.** Contrary to [7, 31], which are limited to a single iteration of editing, our approach facilitates repeated modifications of the same image without departing from the $\mathbf{W}^+$ space. The results of these successive edits are displayed in Fig. 10. The results of multiple editing not only do not distort the image, but also successfully transfer the local features of the reference image. This indicates that our method possesses excellent robustness and flexibility.

### 4.4. Discussions

**Limitations.** Although our optimized editing approach enhances resilience to pose variations, it remains ineffective against substantial pose changes.

**Social impacts.** This technique can benefit sectors like cosmetic surgery, privacy, and design, but it also risks misuse in fabricating misleading information through unauthorized facial alterations. Adhering to legal and ethical standards is essential to prevent such misuse.

## 5. Conclusion

This paper introduces a novel approach to image editing, semantic-aware local editing across images using a single mask. We propose Adaptive Paste-GAN, which detects the optimal region of interest through deformable feature matching, eliminating the need for precise segmentation or annotation, thereby making our method broadly applicable across various object categories. Additionally, our optimization-based strategy effectively transfers stylistic elements from

Figure 8. Local style interpolation. Throughout the optimization process of latent codes, we observe the image's progressive transformation. During this transformation, the alterations to other regions are virtually imperceptible, highlighting the localized nature of the optimization trajectory.

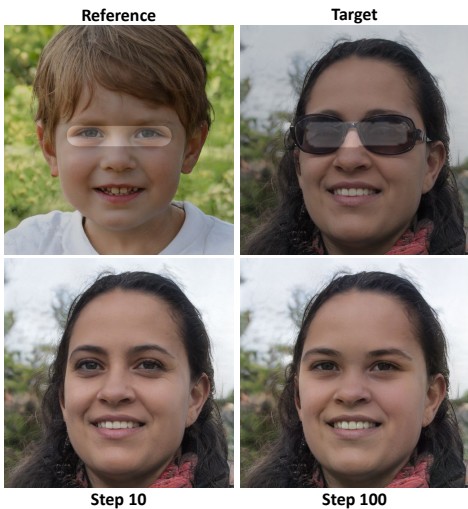

Figure 9. Attribute decoupling. Local editing with varying step numbers shows our method's ability to separate the attributes of eyes and glasses. Initially, the glasses are removed while the eyes remain unchanged. Later, the eyes transform to match the reference images.

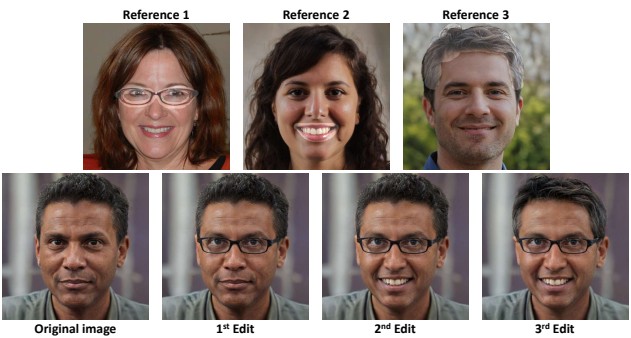

Figure 10. Sequential editing performed on the same target image using three distinct reference images. It demonstrates that the image remains distortion-free, and local features are accurately transferred.

the reference ROI to the target image by manipulating the StyleGAN feature space, achieving a balance between localized edits and global coherence.

Experiments across diverse image categories demonstrate our method's superiority in aligning with user intent. Notably, experiments on local style interpolation and attribute decoupling highlight the high semantic-awareness of our approach.

In the future, we plan to explore the potential of our method in 3D GANs [4, 21], specifically addressing the challenges of cross-pose scenarios.

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
