# Semantic-Aware Local Image Editing with a Single Mask Operation

## Supplementary Material

## A. Comparison with Baselines

To assess our method's performance with arbitrary masks, Fig. 1 presents a comparative analysis between our approach and the feature blending technique proposed by Suzuki et al.[4], as well as BlendNeRF [1]. When users create masks arbitrarily, our model excels at preserving the authenticity and logical consistency of images, enabling edits at a semantic level. In contrast, the feature blending technique is limited to spatial modifications, often resulting in less natural outcomes. While BlendNeRF has some capability for semantic-level editing, it struggles to align with user intent when arbitrary masks are used. Kim et al.[1] suggested using poisson blending[2] to enhance image blending, but our experiments show that this method is unsuitable for arbitrary masks, leading to highly discordant results.

## B. Editing Across Real Images

Through the use of Pivotal Tuning Inversion (PTI) [3], we invert real images into the $\mathbf{W}^+$ space, as illustrated in Fig. 2. Our results demonstrate the capability of our method to conduct local edits among various real images without compromising their authenticity. Such an attribute significantly broadens the utility of our approach, finding extensive applications in fields like medical cosmetology, forensic artistry, and beyond.

## C. Additional Experimental Results

We present an extensive array of editing outcomes within Fig. 3 and Fig. 4, spanning various datasets. The second column shows the results of ROI detection, and the third column shows the results of local editing. These illustrations affirm the versatility of our approach, capable of accommodating diverse applications through user-defined masks across images of numerous object categories.

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

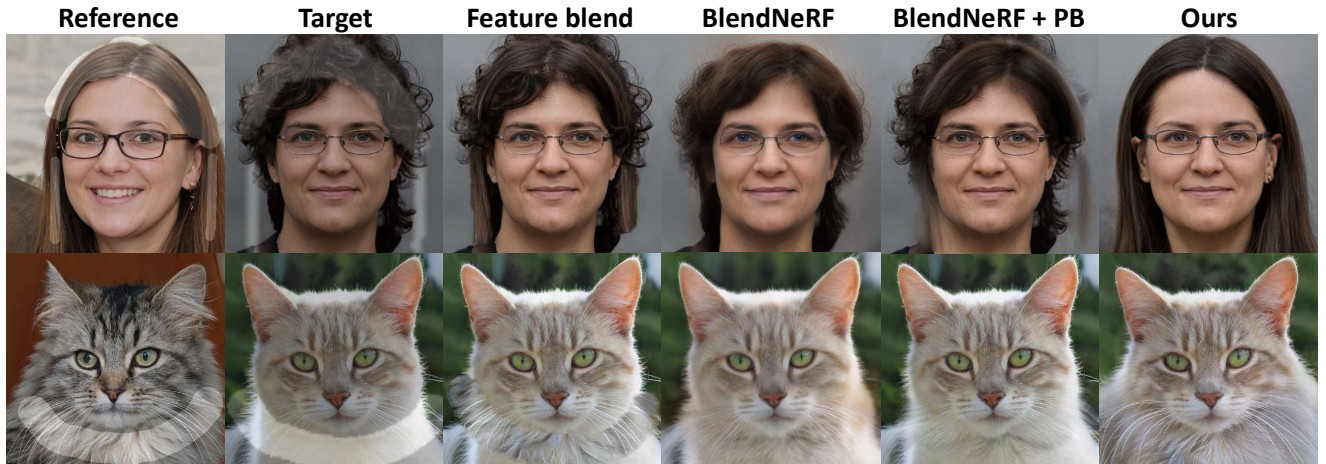

Figure 1. Comparison our method with feature blending and BlendNeRF [1]. Note that PB refers to poisson blending.

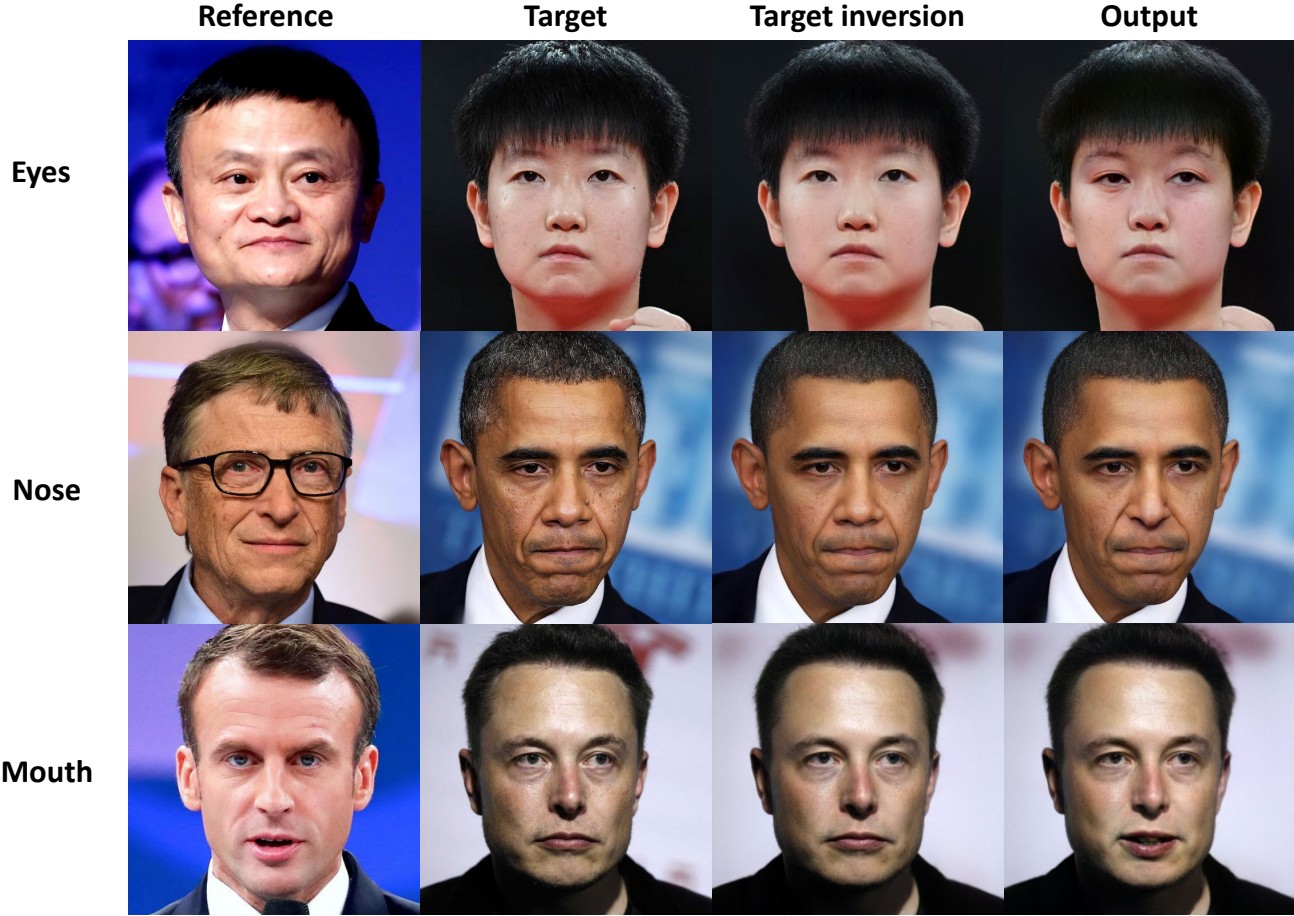

Figure 2. Editing cross real images.

| Reference | Target | Output |
|---|---|---|

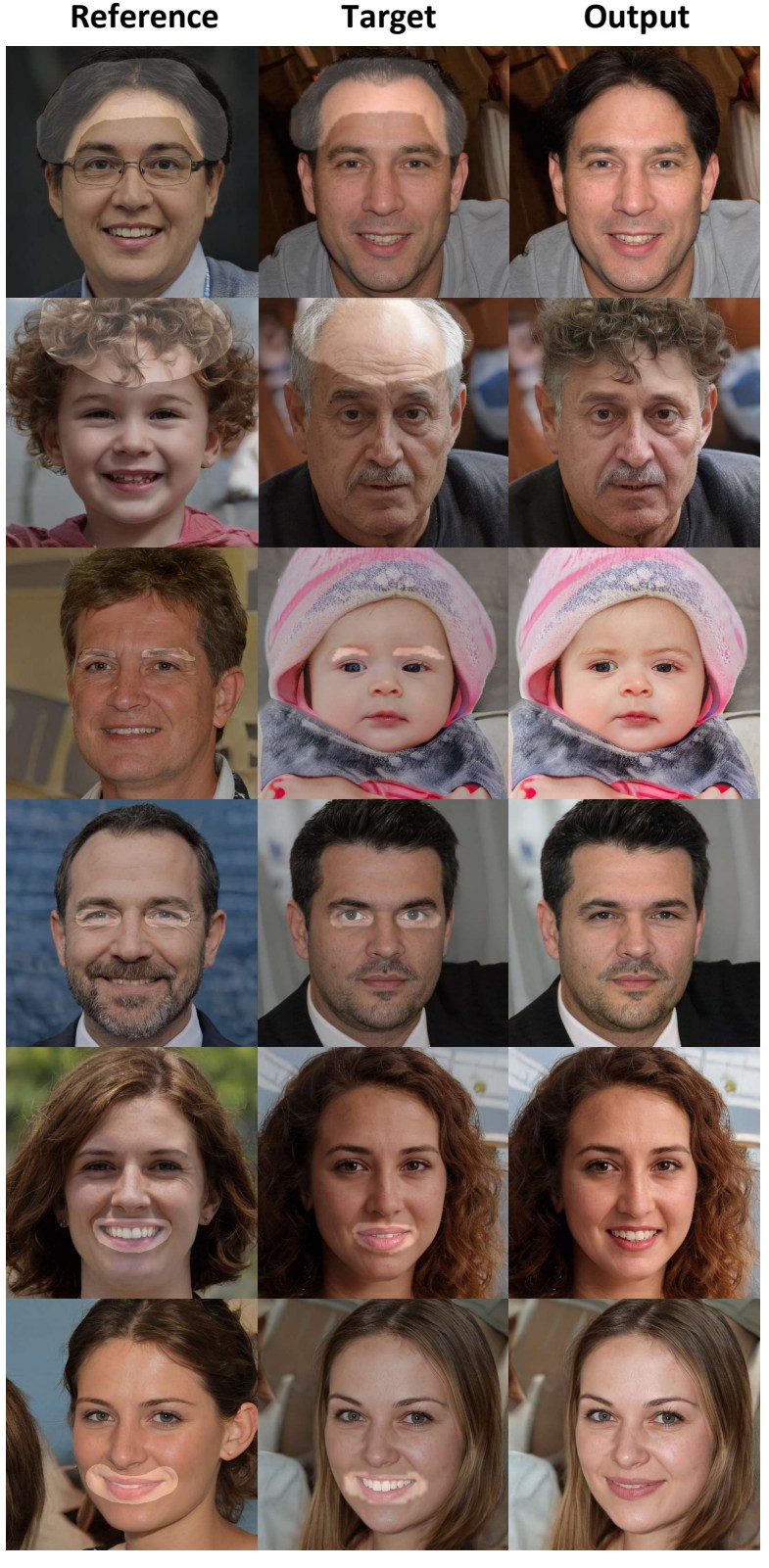

Figure 3. Results on StyleGAN2 trained with the FFHQ dataset

**Reference**     **Target**     **Output**

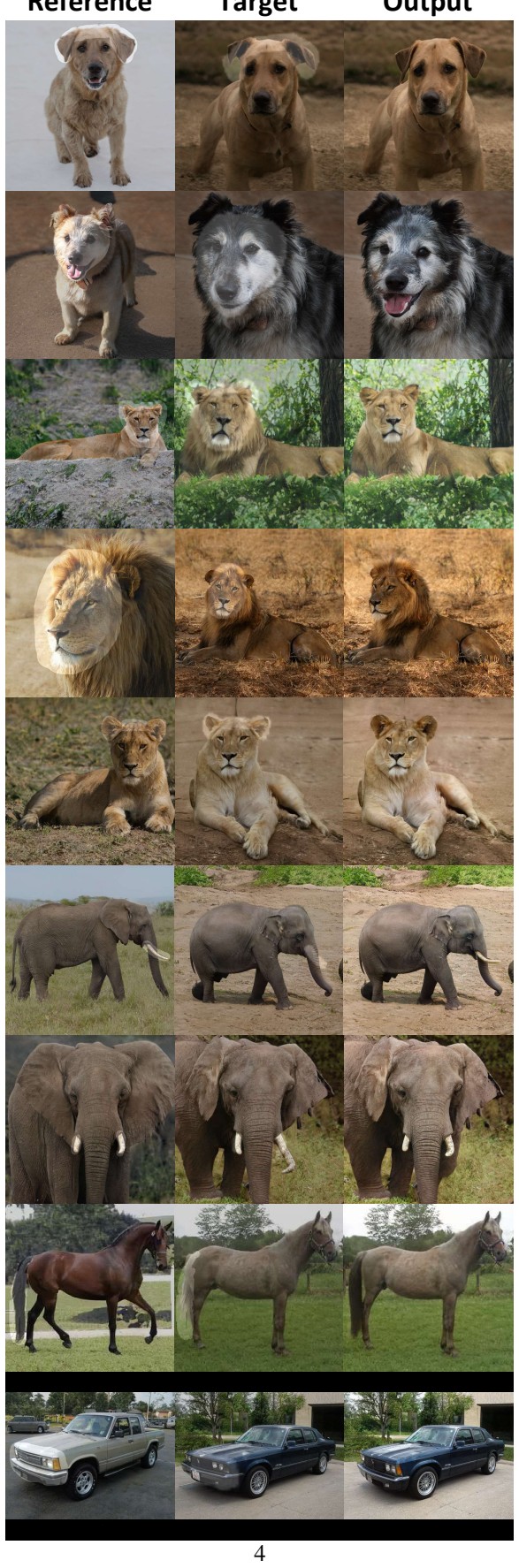

Figure 4. Results on various object categories