# OpenReview forum: "Semantic-Aware Local Image Editing with a Single Mask Operation"
_thecvf.com/CVPR/2025/Workshop/CVEU — CVPR 2025_

### Official Review · Reviewer_SQCn · 2025-03-13

**Rating:** 3
**Confidence:** 3

**Review:**

Summary:

The paper proposes a localized image editing method called Adaptive PasteGAN. It consists of two steps: first, the model detects the editing regions in the image feature map according to user input. Then, a GAN-based generator synthesizes the target semantics while ensuring consistency with the original image. These two stages are performed through test-time optimization and do not require any additional training. The proposed method is evaluated through qualitative comparisons on human face and object-related editing tasks, and shows advantageous performance.

Strengths:

* Formulating editing as a test-time optimization is intriguing and does not require additional training.
* The qualitative results look quite good in terms of naturalness and editing quality compared to baselines including Editing-in-Style and BlendNeRF.

Weaknesses:

* Lack of quantitative comparisons. The authors should evaluate their method on a benchmark dataset and compute quantitative metrics such as editing quality and consistency. This would better justify the benefits of their proposed method.
* There is no time complexity comparison. The paper only mentions that the optimization is typically between 3 seconds and tens of seconds, which is vague. It would be better to have a concrete comparison to better understand the advantages or limitations of the method.
* The proposed deformable feature matching is limited in rotation and scaling, which may limit its flexibility in more complex 3D scenarios, e.g. buildings and non-convex objects.

---

### Official Review · Reviewer_tLqu · 2025-03-16
**This paper presents Adaptive Paste-GAN, a new method for localized, semantic-aware image editing using only a rough mask on the reference image. It does not require additional training, and performs two main steps:  ROI detection using deformable feature matching Latent code optimization for local editing within StyleGAN The method is impressive in how it automatically identifies and aligns corresponding regions between reference and target images, and then seamlessly transfers stylistic features (e.g., hair, eyes, or beards) without affecting unrelated areas.**

**Rating:** 5
**Confidence:** 4

**Review:**

1. Quality
- The method is well-motivated and clearly described.
 - Experiments are thorough, covering various image categories (e.g., FFHQ, LSUN Car, Dog, Cat).
 - Quantitative evaluation (e.g., landmark error for ROI detection) and qualitative comparisons (against Feature Blend, Editing-in-Style, BlendNeRF) are appropriate and convincing.
 - The approach is practical — it works in real-time, supports iterative edits, and remains robust over multiple edits.

2. Clarity
- The paper is well-written, with clear explanations of the method and its components.
- Diagrams (e.g., Fig. 1 and Fig. 3) help visualize the pipeline and the feature alignment process.
- Examples in Figures 4–10 illustrate the editing quality, region matching, and semantic interpolation clearly.

3. Originality
- The idea of combining deformable feature matching with latent code optimization for StyleGAN-based editing is novel.
Unlike prior methods, this approach does not require dual masks, annotated segmentation, or retraining.
- The ROI detection via optimized geometric transformations is an innovative touch.

4. Significance
- This work has strong practical significance for applications like portrait editing, virtual try-on, and artistic content creation.
- It can be integrated into interactive tools due to its speed and flexibility.
- It is likely to influence future work on controllable and user-friendly GAN-based editing.

---

### Official Review · Reviewer_5hfk · 2025-03-17
**Interesting Approach to Local Image Editing with Missing Comparisons**

**Rating:** 4
**Confidence:** 2

**Review:**

This paper introduces Adaptive Paste-GAN, a method for controllable image editing. The proposed approach enables users to apply localized semantic modifications by drawing a rough mask on a reference image, transferring stylistic features to a target image in a semantically aware manner. The method consists of two stages: ROI (Region of Interest) detection using deformable feature matching and local editing via latent space optimization within StyleGAN. The framework aims to provide high-quality, localized edits while maintaining global consistency without requiring additional model training.

**Strengths**

1.	The proposed Adaptive Paste-GAN eliminates the need for precise segmentation or annotations, making it more practical for end users.
2.	Despite being an optimization-based method, the approach maintains high efficiency, enabling near-real-time editing by removing the need for fine-tuning.

**Weaknesses**

1.	The baselines chosen focus primarily on GAN-based methods. Including comparisons with diffusion-based image editing, such as DiffEdit, would provide more up-to-date baselines.
2.	While the paper presents some qualitative results, it would be nice to see some quantitative results to compare the performance of the proposed method against other baselines. A simple example might be comparing CLIP embeddings of reference and edited images.

---

### Decision · Program_Chairs · 2025-03-25

**Decision:**

Accept

**Comment:**

The proposed method is highly practical, user-friendly, and efficient due to its test-time optimization strategy, providing near-real-time localized editing capabilities. The qualitative results convincingly demonstrate superior editing quality and semantic consistency compared to existing baselines, across various image categories.

The primary weakness is the lack of comprehensive quantitative evaluation, including metrics such as CLIP embedding similarity or editing quality benchmarks. Additionally, comparisons with modern diffusion-based editing methods (e.g., DiffEdit) and detailed runtime complexity analysis are missing, which could further validate the method's advantages and limitations.

Given the overall positive evaluations emphasizing the method's novelty, practicality, and strong qualitative results, the paper is accepted. However, authors are strongly advised to address reviewers' concerns by including quantitative evaluations, broader baseline comparisons, and a clear runtime complexity analysis in the camera-ready version.